# Time-Course of Salivary Metabolomic Profiles during Radiation Therapy for Head and Neck Cancer

**DOI:** 10.3390/jcm10122631

**Published:** 2021-06-15

**Authors:** Wakako Yatsuoka, Takao Ueno, Kanako Miyano, Ayame Enomoto, Sana Ota, Masahiro Sugimoto, Yasuhito Uezono

**Affiliations:** 1Dental Division, National Cancer Center Hospital, Tokyo 104-0045, Japan; taueno@ncc.go.jp; 2Division of Cancer Pathophysiology, National Cancer Center Research Institute, Tokyo 104-0045, Japan; kmiyano@ncc.go.jp (K.M.); yuezono@east.ncc.go.jp (Y.U.); 3Institute for Advanced Biosciences, Keio University, Yamagata 997-0052, Japan; ayame.e@ttct.keio.ac.jp (A.E.); sana.ota@ttck.feio.ac.jp (S.O.); msugi@sfc.keio.ac.jp (M.S.); 4Research and Development Center for Minimally Invasive Therapies, Institute for Medical Sciences, Tokyo Medical University, Tokyo 160-8402, Japan; 5Supportive and Palliative Care Research Support Office, National Cancer Center Hospital East, Chiba 277-8577, Japan; 6Department of Pain Control Research, The Jikei University School of Medicine, Tokyo 105-8461, Japan

**Keywords:** metabolomic analysis, oral mucositis, radiation therapy, capillary electrophoresis-mass spectrometry, saliva

## Abstract

Oral mucositis (OM) is one of the most frequently observed adverse oral events in radiation therapy for patients with head and neck cancer. Thus, objective evaluation of OM severity is needed for early and timely intervention. Here, we analyzed the time-course of salivary metabolomic profiles during the radiation therapy. The severity of OM (National Cancer Institute (NCI) Common Terminology Criteria for Adverse Events v3.0) of nine patients with head and neck cancer was evaluated. Partial least squares regression-discriminant analysis, using samples collected before radiation therapy, showed that histidine and tyrosine highly discriminated high-grade OM from low-grade OM before the start of radiation therapy (significant difference, *p* = 0.048 for both metabolites). Further, the pretreatment concentrations of gamma-aminobutyric acid and 2-aminobutyric acids were higher in the high-grade OM group. Although further validations are still necessary, this study showed potentially associated metabolites with worse radiotherapy-related OM among patients with head and neck cancer.

## 1. Introduction

Oral mucositis (OM) is one of the most frequent adverse events in cancer treatment. OM can worsen the patient’s general condition because it reduces oral intake, leads to undernutrition and dehydration, and lowers the patient’s quality of life (QOL) [1,2]. Importantly, OM becomes a portal of infection and triggers a systemic infection that interferes with cancer treatment and affects prognosis and treatment results [3,4]. In turn, the medical costs are increased because of longer hospital stays of patients with infectious diseases. Despite this significant impact, no preventive, therapeutic, or predictive methods for OM have been established to date.

OM occurs in as many as 95% of patients with head and neck cancer who undergo multimodal treatment that includes radiation therapy. It also occurs in as many as 50% of patients treated with radiation therapy alone. In some cases, if radiation therapy is inter-rupted due to OM, it may be necessary to increase the amount of one dose or to supplement with additional radiation, increasing the patients’ burdens [5]. Adoption of intensity-modulated RT (IMRT), use of multimodality imaging to define tumor volume and risk or depict organs [6], and assessments of accuracy of dose distribution using cone-beam computed tomography (CBCT) [7] can mitigate adverse events. However, OM cannot be completely prevented. In general, radiotherapy-related OM resolves in about 10 days and heals in about 1 month, and this healing time tends to be longer than that of OM caused by cell-killing anticancer drugs and molecular target drugs. OM due to chemotherapy or radiation therapy does not result from external stimulation of the mucosa, but rather direct DNA damage to the basal cell layer by reactive oxygen species generated inside the cell, and various cytokines induced by active enzymes. In addition to the direct effect of radiation, OM associated with radiation therapy for head and neck cancer is caused by a decrease in the protective effect of the oral mucosa due to a decrease in saliva secretion, and high susceptibility to infection due to bone marrow suppression, in addition to the direct effect of radiation. OM is an adverse event that is difficult to prevent or treat because of the combined effects of the condition [8,9]. Therefore, patients are at risk of deterioration of QOL and infection for a long period. For example, a rare oral OM called plasma cell mucositis has been reported, and it is said that steroids are effective [10]. In contrast, there is currently no reliable treatment for OM caused by radiation therapy.

Saliva has applications in the diagnosis of various systemic diseases and has the advantage of being less invasive and less costly to collect than other body fluids [11]. We previously isolated candidate substances specific to medication-related osteonecrosis of the jaw (MRONJ) in metabolomic analysis of saliva of patients to identify substances potentially capable of detecting MRONJ onset [12]. Saliva can be readily collected and reflects various pathological conditions, making it an attractive source for diagnosis of systemic diseases, such as oral, breast, and pancreatic cancers [13,14,15]. In addition, serum-based metabolome analysis shows changes in the metabolome before and after radiation therapy for prostate cancer, indicating its potential as a biomarker for personalized treatment [16]. There are also reports of analysis of semen headspace to characterize the patterns of volatile organic compounds (VOCs) in human semen and identify specific metabolites that may be potential biomarkers of pathological conditions [17]. In addition, metabolome analysis is widely used for searching for markers of rejection by T lymphocytes after renal transplantation [18], drug discovery, and personalized medicine [19]. Given the marked impact of OM on the overall well-being of cancer patients, this study aimed to explore biomarkers correlated with the grading score of OM among patients with head and neck cancer who have the highest risk of oral complications during radiotherapy. If the exacerbating factor of OM can be predicted, this will contribute to the prevention of exacerbation by making the intervention more careful. Towards this goal, saliva samples were collected and subjected to metabolome analysis to identify salivary metabolite markers that predict the aggravation of OM at an early stage.

## 2. Materials and Methods

### 2.1. Study Design and Patients

This observational prospective study enrolled 9 male patients (mean age, 63.4 ± 9.81 years) (Table 1) with head and neck cancer and who underwent radiation therapy at the National Cancer Center Central Hospital between 2016 and 2017. Eligibility criteria were: (1) age 20 to 80 years, (2) Performance Status (PS) (Eastern Cooperative Oncology Group (ECOG)): 0–2, (3) patients with head and neck cancer who were scheduled to undergo radiation therapy in which the oral cavity was irradiated with 50 Gy or more, regardless of the combination of chemotherapy and molecular target drug, (4) written consent from the patient was obtained. Exclusion criteria were: (1) patients with a mental illness or psychiatric symptoms and that find it difficult to participate in the study and (2) patients judged by the doctor in charge to be inappropriate for registration in this study, such as those who appear to have strong anxiety regarding their illness or radiation therapy, and those who may have difficulty collecting specimens accurately. The highest grade of OM during radiation therapy was evaluated objectively using the Common Terminology Criteria for Adverse Events (CTCAE) v. 3.0 [20]. Medical examinations including cleaning status were also performed. Grade 2 and above were classified with the high-grade group. The number of people as a factor that weakens and damages the oral mucosa is shown in each of the two groups (Table 2).

### 2.2. Sample Collection

Saliva samples were collected four times: (1) before the start of radiation therapy (from the first dental examination to the day before the start of irradiation), during radiation therapy when the irradiation dose was (2) 20 Gy ± 2 days (first and second sessions; 8 days apart) and (3) 40 Gy ± 2 days (second and third sessions; 6 days apart), and (4) within 1 week after the completion of radiation therapy (Figure 1).

Saliva was collected following the protocols described in our previous study [12]. Briefly, unstimulated whole saliva was collected using a polypropylene tube (50-cc Falcon tube; Corning Inc., Corning, NY, USA) on ice. Patients were required to undergo a 12-h fast (no food after 9:00 p.m. the day before sample collection), and saliva was collected the following morning. The patients were also prohibited from drinking, smoking, or using oral hygiene products for at least 1 h before saliva collection. They were instructed to rinse their mouth with tap water and to spit into the collection tube after 5 min. The tube was placed in a Styrofoam cup filled with crushed ice in water. The patients were also instructed not to cough up mucus. The amount of saliva collected ranged from 0.1 mL to 0.5 mL The collected saliva was transferred within 1 h to a freezer at −80 °C until use.

### 2.3. Metabolomic Analysis

The processing protocol for saliva samples has been described elsewhere [21]. Briefly, frozen saliva was thawed at 4 °C for 1.5 h. The samples were then filtered through a 5-kDa cutoff filter (Millipore, Billerica, MA, USA) via centrifugation at 9100× *g* for at least 2.5 h at 4 °C. The filtrate (45 μL) was transferred to a 1.5-mL Eppendorf tube, and 5 μL of water containing 2 mM methionine sulfone, 2-[N-morpholino]-ethanesulfonic acid, d-camphol-10-sulfonic acid, sodium salt, 3-aminopyrrolidine, and trimesate was added. The mixture was mixed for 30 s using a vortex mixer and centrifuged at 4550× *g* for 1 min at 4 °C. Then, 7 μL was transferred to a vial for capillary electrophoresis (CE) time-of-flight (TOF) mass spectrometry (MS) analysis. The instruments and parameters of CE-TOFMS have been described previously [21].

We analyzed CE-TOF-MS raw data using our proprietary software, MasterHands (Keio University, Yamagata, Japan) [13]. The peaks were identified by matching the corresponding *m**/z* values and normalized migration times to those of standard compounds. The absolute concentration of each metabolite was calculated based on the peak areas of the metabolite in each sample, and the corresponding standard compounds in the mixture with internal standards. All samples were measured in a single batch to eliminate unexpected bias.

### 2.4. Statistical Analysis

The patients were classified into low-grade (0 and 1) and high-grade (2) groups based on mucositis severity. Between-group differences were analyzed using partial least squares regression-discriminant analysis (PLS-DA) in MetaboAnalyst [22]. PLS-DA used quantified metabolite concentration data by repeated dimensional reduction and regression to explain low- and high-grade groups [23]. The quantified metabolites in the samples collected before radiation therapy were used for this analysis. Metabolites showing ≥1.5 or <0.5-fold change of averaged value between low- and high-grade groups were used. Since PLS-DA is a supervised method, cross-validation was used to access the generalization ability [24].

To evaluate the discrimination ability of an individual metabolite, receiver operating characteristic (ROC) analyses were conducted. The higher-grade group was discriminated by the lower-grade group by changing the threshold, and sensitivity and specificity were calculated using ROC curves.

The Mann–Whitney test was used for the evaluation of the differences in the quantified metabolites between groups. All statistical analyses were performed using the GraphPad Prism software (v.6.0; GraphPad Software, San Diego, CA, USA) and BellCurve for Excel (Social Survey Research Information Co., Ltd., Tokyo, Japan). Differences with a *p* < 0.05 were considered statically significant.

## 3. Results

### 3.1. Patient Characteristics

Good plaque control and no tongue coating or white moss were defined as good oral cleaning conditions. Regarding oral cleaning condition, two out of three patients in the high-grade group and three out of six in the low-grade group had good oral condition.

### 3.2. Profiled Metabolites

Metabolomic analysis using CE-TOFMS successfully identified 214 metabolites. Of these, 71 metabolites were detected ≥ in 80% of the samples. PLS-DA using the sample collected before radiation therapy showed a difference in metabolomic profiles between the low-grade and high-grade groups (Figure 2). Score plots revealed that six samples from the low-grade group (green plots) have relatively higher values in both the first and second components (appearing in the upper right area). Meanwhile, three samples in the high-grade group (pink plots) showed relatively lower first components (appearing in the left area) (Figure 2A).

For variable importance in projection (VIP) scores, metabolites with higher VIP helped discriminate between low- and high-grade OM, assessed by PLS-DA (Figure 2B). Various amino acids showed higher VIP values; these included histidine (His), tyrosine (Tyr), glycine (Gly), glutamic acid (Glu), aspartic acid (Asp), tryptophan (Typ), lysine (Lys), and methionine (Met). The Mann–Whitney test was also used for evaluating the discriminations of each metabolite. Of these, His and Tyr showed a significant difference (*p* = 0.048 for both metabolites). Urocanate, an intermediate metabolite in histidine metabolites, also showed a higher VIP score. Other significant metabolites, gamma-aminobutyric acid (GABA), butyrate, 2-isopropaylate, and 2-aminobutyric acids, were also identified.

The receiver operating characteristic (ROC) curves of His and Tyr concentrations in the saliva collected before radiation therapy are depicted in Figure 3. Both metabolites resulted in an area under the ROC curve (AUC) of 0.94 (95% confidence interval (CI): 0.79−1.0). The time courses of metabolite concentrations showing a significant difference (*p* < 0.05, Mann–Whitney test) between the high-grade and low-grade groups in at least onetime-point are depicted in Figure 4. The concentrations of all 10 metabolites were higher in the high-grade group than those in the low-grade group across all the four time-points of collection. His and Tyr showed the same tendency in the analysis in which the irradiation field was unified in the oropharynx and oral cavity (*n* = 4, No.3, 4, 5, 9) in Figure 5.

## 4. Discussion

The adverse impact of OM on both treatment and patient outcomes among those with head and neck cancer who undergo radiation therapy highlights the need for biomarkers that predict its severity. This study found that salivary metabolome analysis can be used to identify metabolites associated with worse radiation therapy-related OM among patients with head and neck cancer, and these metabolites can be used to predict its severity.

Several risk factors for the exacerbation of OM, including the type of chemotherapy regimen, radiation dose, inadequate oral cleansing status, and history of smoking, have been established. Although the number of cases was small, as a characteristic of patients regarding exacerbation of OM, diabetes was not in the high-grade group, and two out of three patients had good oral cleaning condition. Since the exacerbation factors involve complicated factors, it is considered effective to investigate from a more detailed viewpoint by metabolomics analysis. However, to our best knowledge, there are no reports on the use of microbial metabolites measured in real-time for predicting the risk and severity of OM. We found higher salivary GABA and 2AB concentrations before radiation therapy in the high-grade group. GABA is an amino acid derivative that functions as an inhibitory neurotransmitter. It can increase without being decomposed into succinic acid due to several reasons, including stress. Originally, it enters the tricarboxylic acid (TCA) cycle, and its inability to do so is speculated to cause abnormalities in the TCA cycle. This in turn causes failure of mucosal repair due to mitochondrial dysfunction. Metabolome analysis of blood samples collected from patients with depression has also shown that the GABA concentration correlates with the severity of depression [25]. Further, the immune function of phytohemagglutinin (PHA), a type of lectin, is reduced in depressive states [26]. This may be associated with the aggravation of mucositis in patients under high stress. It has been reported that the patients with cancer received a DSM-III diagnosis, with 44% being diagnosed as manifesting a clinical syndrome. Approximately 68% of the psychiatric diagnoses consisted of adjustment disorders, with 13% representing depression [27]. The plasma GABA concentration is increased in stomatitis and disturbed sleep conditions. Chen et al. reported that the administration of Kouyanqing granules, a Chinese herbal medicine, improved stomatitis and decreased GABA concentration in plasma samples [28].

We also performed metabolome analysis in plasma and found changes in tryptophan metabolism, D-glucose, and D-glutamate, which may all affect the TCA cycle. 2AB increases the detoxification and synthesis of glutathione, an antioxidant. Under high oxidative stress conditions, glutathione is consumed and is increased to supplement 2AB; thus, it may indicate the conditions in which mucositis is more likely to become severe. However, to our knowledge, no previous study has shown that the concentrations of GABA and 2AB are increased in oral cancer.

The current study found metabolites with high VIP scores and higher concentrations before radiation therapy in the low-grade mucositis group. Of these, histidine, which is known to have a repairing and an antioxidant effect on the intestinal mucosal epithelium [29], showed the highest VIP value. This metabolite is a precursor of carnosine, which also exerts an antioxidant effect by removing free radicals [29,30]. In addition to histidine, concentrations of tyrosine were also high during and after radiation therapy in the low-grade mucositis group. Ornithine has been reported to act as an anti-stress agent [31].

Glutamine is converted to glutamic acid in the mitochondria. It has been reported that glutamine administration during radiation therapy for head and neck cancer may markedly lower the incidence and severity of OM [32,33,34]. Glycine is a precursor of glutathione, which has antioxidant properties. Tyrosine supplementation has been shown to reduce fatigue under stress and improve mood and cognitive function [35]. Accordingly, these metabolites may be used to predict the severity of OM.

In this study, spacers were used in all cases; thus, metal scattering rays had only a minor influence on the exacerbation of OM [36]. In one patient who underwent radiotherapy alone, the target organ was only the parotid gland, but the patient still developed high-grade OM. Treatment response varies between patients and between chemotherapeutics, and this particular case shows that the radiation field alone may not be able to predict the severity of mucositis. In this regard, metabolome analysis may be useful for the early identification of patients at risk of high-grade OM.

This study has some limitations. Epidemiologically, it is said that the proportion of males in oral cancer is higher than that of females, but the overall number of cases in this study was small, and only males participated this time. The number of cases was small, and treatment factors, including the irradiation dose and type of chemotherapy, were not standardized. Patients who receive radiation therapy, including in the oropharynx and oral cavity, have a high frequency of oral OM; however, regarding oral pain felt compared to patients receiving radiation therapy in the hypopharynx, there is a report that it does not change [37]. Although the number of cases is small, His and Tyr showed the same tendency in the analysis in which the irradiation field was unified in the oropharynx and oral cavity. When comparing radiation therapy alone with cisplatin or cetuximab, the incidence of OM is 10 to 20 times higher [38]. Additionally, there is little difference in the incidence of OM of Grade 3 or higher between chemoradiation therapy alone or in combination with cetuximab for oropharyngeal, hypopharyngeal, and laryngeal cancers [39]. However, these studies on the incidence of OM involved different target diseases and irradiation doses. This is the first study to use metabolome analysis to study OM. Compared with recent reports using metabolomic analysis, such as those using 12 patients with Sjögren’s syndrome [40], the number of specimens used here (*n* = 9) is comparable. It is said that the time required to cure OM associated with radiation therapy tends to be longer than that of cell-killing anticancer drugs. In this study, the specimen was sampled within 1 week after radiation therapy. However, we will consider collecting specimens for a longer period in the future. In this study, grade ≥2 was classified into the high-grade group. Grade 2 is also painful and was considered clinically important because dietary effects impair immunity, infect ulcers, and delay the healing of mucositis. Even in Grade 2, analgesics may be used for pain and the drug may cause side effects. In the future, when increasing the number of cases, we would like to examine whether the results obtained in this study tend to be stronger at grade ≥3. Further studies should be conducted using a large number of cohorts, a standardized treatment protocol, and an accurate dose-volume histogram (DVH), and by creating a cohort that unifies the target diseases.

## 5. Conclusions

Salivary metabolome analysis identifies metabolites associated with the severity of OM to identify patients who are at high risk of severe OM among those with head and neck cancer undergoing radiation therapy. However, large-scale validation is necessary considering the heterogeneity of the response and background of each patient.

## Figures and Tables

**Figure 1 jcm-10-02631-f001:**
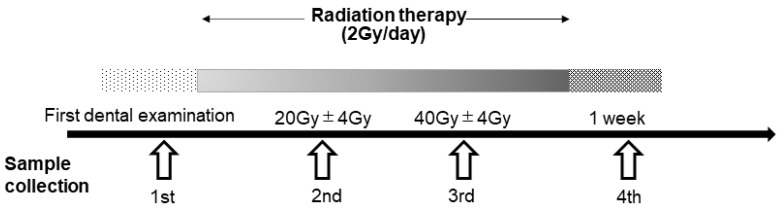
Flowchart of sample collection during radiation therapy. Saliva samples were collected at the following four time-points. (1) At the start of radiation therapy (from the first dental examination to the day before the start of irradiation), (2) during radiation therapy (irradiation dose of 20 Gy ± 2 days, first and second sessions; 8 days apart), (3) during radiation therapy (irradiation dose of 40 Gy ± 2 days, second and third sessions; 6 days apart) and (4) after completion of radiation therapy (from the end date to 1 week after the end).

**Figure 2 jcm-10-02631-f002:**
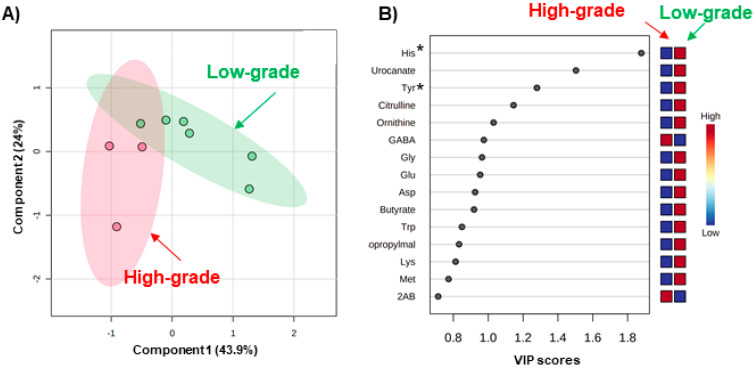
PLS-DA of metabolite concentrations in the saliva samples collected before radiation therapy. (**A**) Score plots. The predictive capabilities are *R*^2^ = 0.93 and *Q*^2^ = 0.40 by 9-fold-cross validation. The pink and green plots indicate the patients with high- and low-grade oral mucositis. The plots with short distances indicate a high similarity pattern of salivary metabolite concentrations, and, oppositely, the plots with large distances indicate large different patterns. The filled circles colored light pink and light green indicate the 95% confidence intervals of each group. For these analyses, the absolute metabolite concentration was normalized using scale range. X and Y axes are the 1st and 2nd components, e.g., the most and second most important components to explain the high- and low-grade groups. Their cumulative contribution rates are 43.9% and 24%, respectively (**B**) VIP score. Each plot indicates the VIP score of the metabolites used in this analysis. The higher VIP score indicates a higher contribution to discriminate the high-grade and low-grade groups. * is added to the metabolites with *p* < 0.05 (Mann–Whitney test). Boxes on the right indicate a heatmap. Red and blue indicated higher and lower concentrations. For example, regarding histidine (His), the averaged concentrations were lower in the high-grade groups compared to those in low-grade groups.

**Figure 3 jcm-10-02631-f003:**
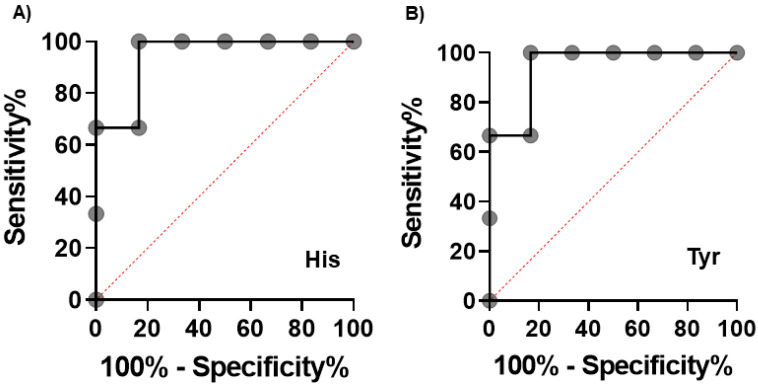
ROC curves to discriminate higher grade group (*n* = 3) from lower grade group (*n* = 6) based on the saliva collected before radiation therapy. The X and Y axes indicate the 100-specificity (%) and sensitivity (%), respectively. (**A**) His and (**B**) Tyr. The larger area under the curves indicates higher sensitivity of accuracy to discriminate. The AUC values were 0.94 (95% confidence interval (CI): 0.79−1.0, *p* = 0.039) for both metabolites.

**Figure 4 jcm-10-02631-f004:**
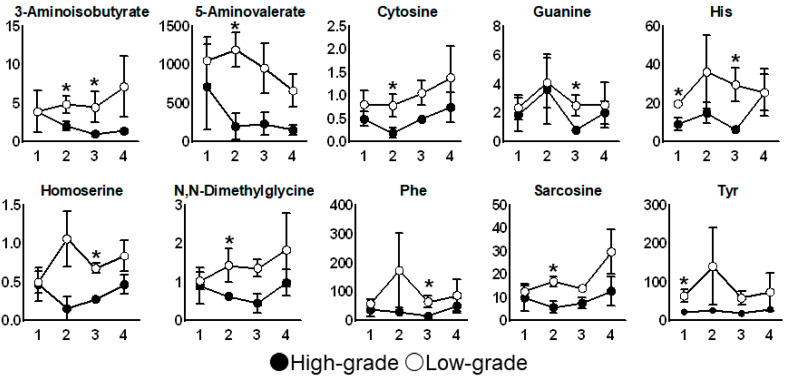
Time course of metabolites showing significant differences in their concentration (Mann–Whitney test) at at least one time-point between the low- (open circle) and high-grade (filled circle) oral mucositis group. The mean and standard errors are shown. * *p* < 0.05. The *x*-axis indicates the sampling point described in Figure 1. The *y*-axis indicates the concentration of each metabolite (μM).

**Figure 5 jcm-10-02631-f005:**
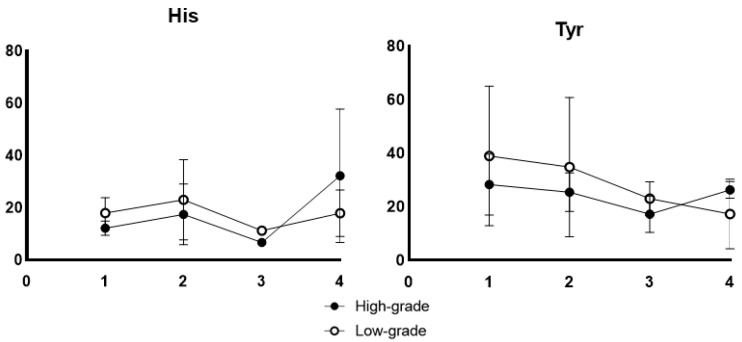
Time course of His and Tyr showed the same tendency in the analysis in which the irradiation field was unified in the oropharynx and oral cavity. (*n* = 4, No.3, 4, 5, 9)) The *x*-axis indicates the sampling point described in Figure 1. The *y*-axis indicates the concentration of each metabolite (μM).

**Table 1 jcm-10-02631-t001:** Characteristics of the patients with head and neck cancers.

No.	Age (Years)	Sex	Disease	Chemotherapy Regimen	Total Radiation Dose (Gy)	Highest Grade of Mucositis (CTCAE v3)
1	49	male	Parotid gland carcinoma	Not treated	60	2
2	70	male	Hypopharynx carcinoma	CDDP(80 mg/m^2^) 1 Kur	70	0
3	70	male	Oropharynx carcinoma	CDDP(80 mg/m^2^) 2 Kur	70	0
4	73	male	Oropharynx carcinoma	CDDP1 Kur∔Cetuximab 5 Kur	70	2
5	46	male	Tongue carcinoma	CDDP(80 mg/m^2^) 3 Kur	70	2
6	57	male	Hypopharynx carcinoma	CDDP(80 mg/m^2^) 3 Kur	68	1
7	73	male	Hypopharynx carcinoma	Cetuximab 6 Kur	70	1
8	70	male	Hypopharynx carcinoma	CDDP(80 mg/m^2^) 3 Kur	70	0
9	63	male	Maxillary sinus carcinoma	CDDP(80 mg/m^2^) 3 Kur	64	1

**Table 2 jcm-10-02631-t002:** Patient characteristics associated with exacerbation of oral mucositis.

	High-Grade Group (*n* = 3)	Low-Grade Group (*n* = 6)
Smoking	3 (100%)	6 (100%)
Diabetes melitus	0 (0.0%)	2 (33.3%)
Sharp edeges of teeth	0 (0.0%)	0 (0.0%)
Xerostomia	2 (66.6%)	4 (66.6%)
Undernutrition (ALB < 3.5 g/dL)	2 (66.6%)	4 (66.6%)
Good oral cleaning condition	2 (66.6%)	3 (50.0%)
Oral cavity or Oropharynx carcinoma	2 (66.6%)	2 (33.3%)

## Data Availability

The data presented in this study are available on request from the corresponding author.

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
