# Peer review of "Time-Course of Salivary Metabolomic Profiles during Radiation Therapy for Head and Neck Cancer"

_jcm, 2021, doi:10.3390/jcm10122631_

Round 1

Reviewer 1 Report

The revisions are good. 

Reviewer 2 Report

In this paper, the authors aimed to evaluate the utility of salivary metabolites in prediction of oral mucositis (OM) severity of head and neck cancer patients after radiation therapy. Saliva samples were collected from nine head and neck cancer patients before radiation therapy and metabolomic profiles between low- and high-grade om groups were compared. They found that histidine and tyrosine showed high performance for discriminating low- and high-grade OM group, and that concentrations of gamma-aminobutyric acid and 2-aminobutyric acids were higher in the high-grade OM group.

  1. First of all , I do not think the conclusions from a cohort of nine participants are reliable. In addition, the sample selection strategy does not make any senses. Since the authors planned to include only nine participants, they should control as many covariates as possible to minimize the possibility of confounding. However, among the limited characteristics the authors provided in Table 1, the range of age is 49-73 years old with a mean±SD of (63±10), and most importantly, all participants should have a same cancer type, however, five types of head and neck cancer presented in these nine participants. Further, all variables in Table 2 should also be controlled because they are related to OM. Based on these, it is very probable that the results are chance findings.
  2. There are tracked changes in the manuscript, I have reviewed papers from > 20 journals, but this is my first time to see such a manuscript. It is not scientific.
  3. The Methods sections lack details. What normalization method did the author use?
  4. What do the component 1 and 2 represent in Figure 2A? How were they calculated? How did the authors perform ROC analyses? How many samples used?

Reviewer 3 Report

I have some suggestions for the authors:
In the Introduction part add the differences between oral mucositis caused by radiotherapy and the other mucositis such as plasma cell mucositis and lichenoid mucositis, this could help the readers to better understand the differences between this pathologies in terms of etiopathogenis and treatments.
In the Methods part, "Others that the doctor in charge judges to be inappropriate for registration in this study", please specify which the doctor judged inappropriate.
Furthermore, I suggest a review of the English form.

Reviewer 4 Report

First of all, it is the first time that I receive a paper for the revision with comment and corrections of the authors. It is very impossible to read in this form. 

I intend to do a more in-depth review when I am provided with a file suitable for reading.

In general, the paper is interesting but in introduction section is very poor.

The authors must be improved. 

I suggest to read these pubblications to learn more about the part on radiotherapy:

Mazzola R, Fiorentino A, Ricchetti F, Gregucci F, Corradini S, Alongi F. An update on radiation therapy in head and neck cancers. Expert Rev Anticancer Ther. 2018 Apr;18(4):359-364. doi: 10.1080/14737140.2018.1446832.

Fiorentino A, Cozzolino M, Caivano R, Pedicini P, Oliviero C, Chiumento C, Clemente S, Fusco V. Head and neck intensity modulated radiotherapy parotid glands: time of re-planning. Radiol Med. 2014 Mar;119(3):201-7. doi: 10.1007/s11547-013-0326-3.

No mention is made about roule of metabolomics in clinical field. This paper is based on metabolomics but in introduction section,  it is completely absent. I suggest to read these papers:

Longo, V., Forleo, A., Provenzano, S. P., Coppola, L., Zara, V., Ferramosca,
A., et al. (2018). HS-SPME-GC-MS metabolomics
approach for sperm quality evaluation by semen volatile organic
compounds (VOCs) analysis. Biomedical Physics & Engineering
Express, 5, 015006.

Nalbantoglu S, Abu-Asab M, Suy S, Collins S, Amri H. Metabolomics-Based Biosignatures of Prostate Cancer in Patients Following Radiotherapy. OMICS. 2019 Apr;23(4):214-223. doi: 10.1089/omi.2019.0006. PMID: 31009330.

Futhermore, I suggest to explain all abbrevations (for example, in line 67: PS (ECOG)).

I look forward to receiving the work in a definitive form for other considerations and evaluations. 

Round 2

Reviewer 2 Report

I do not think the authors revisions addressed my concerns.

I insist my standing point that the conclusions from a cohort of nine participants, without any attempts to control potential covariates and confounders that are associated with both the exposure and outcome, are reliable. As you can find in the authors’ response, they did not try to increase the sample size, neither did they fix the issue of variable matching.

Reviewer 3 Report

Authors followed the reviewers' suggestions.

In the part of therapeutic opportunities of oral mucositis, I suggest to add this recent reference in the text [PMID: 32148974] because showed an alternative treatment for the management of oral mucositis (line 61).

After this addiction, the paper can be suitable for publication.

Reviewer 4 Report

In introduction section, metabolomic importantance in clinical field is yet not much emphasized. 

From line 63, a discussion that delves into this point is recommended with several examples of use of metabolomic/volatilomic in diagnosi, prognosis or study of side effects.

Figure 1 is low quality. Replace it. 

Author Response

This manuscript is a resubmission of an earlier submission. The following is a list of the peer review reports and author responses from that submission.

Round 1

Reviewer 1 Report

The writing and organization of the manuscript is good. 

A moderate check on the spacing between words is required. 

The figure quality needs to be improved. 

Author Response

The written and organization of the manuscript is good.

A moderate check on the spacing between words is required.

We thank this observation. according to reviewer's comment, we checked the spaces and also revised the English throughout manuscript.

The figure quality needs to be improved.

According to the reviewer's comment, we improved the quality of all figures. We also modified Figure 1.

Reviewer 2 Report

Study group is relatively small, therefore few samples do non allow to make an appropriate conclusions. Number of patients needs to be increased.

Author Response

Comment and Suggestion for Authors

Study group is relatively small, therefore few samples do non allow to make an appropriate conclusions. Number of patients needs to be increased.

We appreciate this important comment. To clarify this limitation, we waken the conclusion and clearly state this problem in the last sentence of the Abstract (Page 1) and the last sentence of the Discussion (Page 7, lines 235-236, page 8, lines 246-249). We also declared this limitation at the Conclusion (page 8, lines 260-264).  

Reviewer 3 Report

Dear Editors,

the manuscript of Yatsuoka et al, tries to identify new Salivary metabolomics to predict oral mucositis during radiation therapy for head and neck cancer. Also the overall topic sound interesting, the manuscript has several weak point.

The study design is not new - investigations over a longer time point would be helpful.

The groups are very small, wich makes it difficult to see significant differences. There are very few data on patients data.

The text is not written clearly and needs improvement.

The figure legend are lacking important informations.

The figures itself are not very clear

Author Response

Open Review 

the manuscript of Yatsuoka et al, tries to identify new Salivary metabolomics to predict oral mucositis during radiation therapy for head and neck cancer. Also the overall topic sound interesting, the manuscript has several weak point.

The study design is not new-investigations over longer time point would be helpful.

We agree with this reviewer's suggestion. We revised the Discussion (page 8, lines 249-252).

The groups are very small, which makes it differences. There are very few data on patients data. 

We thank the reviewer's acknowledging the importance of this study. To clarify this limitation, we weaken the conclusion and clearly state this problem in the last sentence of the Abstract (Page 1) and the last sentence of the Discussion (page 7, lines 235-236 ,page 8, lines 246-249). We also declared this limitation at the Conclusion (page 8, lines 260-264).There was significant concern regarding the saliva collection from the patients with head neck cancer during radiation therapy. However, we confirmed that enough volume of saliva for metabolome analyses can be collected. We add details of the chemotherapy regimen as a patient's data to Table 1.

The text is not written clearly and needs improvement. 

We thank this observation. According to the reviewer's comments, we revised the English throughout the manuscript and also revised the Materials and Methods (page 2, lines 63-64, page 3, lines 83-84 and 86-88) and the Conclusion (page 8, lines 260-264).

The figure legends are lacking important informations.

According to reviewer's comments, we revised the figure legends to provide more details.

The figures it self are not very clear

According to the reviewer's comments, we improved the quality of all figures.

Reviewer 4 Report

Authors performed the salivary metabolome analysis to identifies metabolites associated with radiotherapy-related acute oral mucositis (OM) by using saliva samples of 4-times collection during cancer treatment including radiotherapy. It was interesting, in terms of that this study reported the analysis of saliva metabolites measured in real-time for predicting the risk of OM. However, there are many flaws in current study. First, the number of cohorts was very small (only 9 cases). Furthermore, this small cases had various primary cancer (4 Hypopharynx carcinoma, 2 Oropharynx carcinoma, 1 Salivary gland carcinoma, 1 Oral cavity carcinoma, and 1 Maxillary sinus carcinoma). Patients with oropharynx carcinoma and oral cavity carcinoma, who were prone to OM, accounts for only 30 percent. Next, regarding to the classification of OM group (lower vs. higher grade), higher grade OM was defined CTCAE grade 2 or higher. Considering the following characterization of OM and the clinical condition, grade 3 or higher is considered to be appropriate as a high grade OM. For the characterization of mucositis, CTCAE grades were defined as follows: grade 0: no mucositis; grade 1: asymptomatic or mild symptoms; grade 2: moderate pain, does not interfere with oral intake but modified diet is indicated; grade 3: severe pain, interferes with oral intake; grade 4: life-threatening consequence requiring urgent intervention. grade 5: death It is necessary to re-evaluate the clinical importance of this study as there are no patients with high grade OM if divided into groups in this way, low-grade (0, 1, and 2) vs. high-grade (3 or more) groups. In other words, it is questionable whether the results of this study are of clinical value because they were conducted on patients without severe side effects. Current metabolomics analysis has shown the importance of various amino acids with too few samples. Increasing the number of cases is essential to increase the statistical power of the results. As you know, the type of chemotherapy regimen, radiation dose, inadequate oral cleansing status, and history of smoking, and so on have contributed to OM. Please provide the precise RT planning information for irradiation. including the DVH for the target and normal tissues. And, provide the precise chemotherapy information. In summary, for suggesting metabolites in saliva as a predictor related to OM, it is necessary the increasing number of head and neck patients including oropharynx carcinoma and oral cavity carcinoma patients who are known to have a high incidence of OM, who treated with relatively homogeneous treatment approach.

Author Response

Authors performed the salivary metabolome analysis to identifies metabolites associated with radiotherapy-related acute oral mucositis (OM) by using saliva samples of 4-times collection during cancer treatment including radiotherapy. It was interesting, in terms of that this study reported the analysis of saliva metabolites measured in real-time for predicting the risk of OM. However, there are many flaws in current study. First, the number of cohorts was very small (only 9 cases).

We agree this reviewer's criticism. To clarify this limitation, we weaken the conclusion and clearly state this problem in the last sentence of the the Abstract (page 1) and the last sentence of the Discussion (page 7, lines 235-236 ; page 8, lines 246-249). We also declare this limitation at the Conclusion (page 8, lines 260-264).

Furthermore, this small cases had various primary cancer (4 Hypopharynx carcinoma, 2 Oropharynx carcinoma, 1 Salivary gland carcinoma. 1 Oral cavity carcinoma, and 1 Maxillary sinus carcinoma). Patients with oropharynx carcinoma and oral cavity carcinoma, who were prone to OM, account for only 30 percent,

We thank the reviewer's comment. To clarify, we revised the Discussion (pages 7-8, lines 235-247).

Next, regarding to the classification of OM group(lower vs. higher grade), higher grade OM was defined CTCAE grade 2 or higher. Considering the following characterization of OM and the clinical condition, grade3 or higher is considered to be appropriate as a high grade OM. For the characterization of mucositis, CTCAE grades were defined as follows: grade 0: no mucositis; grade 1: asymptomatic or mild symptoms; grade 2: moderate pain, dose not interferes with oral intake but modified diet is indicated; grade 3: severe pain, interferes with oral intake; grade 4: life-threatening consequence requiring urgent intervention. grade 5: death It is necessary to re-evaluate the clinical importance of this study as there are no patients with high OM if divided into groups in this way, low-grade (0, 1, and 2) vs. high-grade (3 or more) groups. In other words, it is questionable whether the results of this study are of clinical value because they were conduct on patients without severe side effects.

We agree with this reviewer's criticism. We revised the Discussion (page 8, lines 252-255).

Current metabolomics analysis has shown the importance of various amino acids with too few samples. Increasing the number of cases is essential to increase the statistical power of the results.

We agree with the reviewer's comment. We plan to perform a validation study in the future. We revised the Discussion (page 7, lines  235-236, page 8 lines 246-249). We also declared this limitation at the Conclusion (page 7, lines 260-264).

As you know, the type of chemotherapy regimen, radiation dose, inadequate oral cleansing status, and history of smoking, and so on have contributed to OM. Please provide the precise RT planning information for irradiation. including the DVH for the target and normal tissues. And, provide the precise chemotherapy information. In summary, for suggesting metabolites in saliva as a predictor related to OM, it is necessary the increasing number of head and neck patients including oropharynx carcinoma and oral cavity carcinoma patients who are known to have a high incidence of OM, who treated with relatively homogeneous treatment approach.

According to the reviewer's comment, we added details of the chemotherapy regimen to Table 1. As you point out, risk factors for exacerbation of oral mucositis include the type of chemotherapy of chemotherapy regimen, radiation dose, inadequate oral cleaning status, and history of smoking, but there are no reports of real-time measurement of microbial metabolites. In the future, we hope to find new risk factors that may lead to the development of the development of guidelines. To clarify, we revised the Discussion (page 8, lines 256-259).

Round 2

Reviewer 2 Report

Despite the Authors tried clarified limitation of their study, it seems not be solved and improved.

Reviewer 3 Report

the authors did not improve manuscript in its present form. Not even the figures have been improved. Since there were substantial points to address, just deleting parts of the discussion is just not enough.